# Plant-Derivatives Small Molecules with Antibacterial Activity

**DOI:** 10.3390/antibiotics10030231

**Published:** 2021-02-25

**Authors:** Sana Alibi, Dámaso Crespo, Jesús Navas

**Affiliations:** 1Analysis and Process Applied to the Environment UR17ES32, Higher Institute of Applied Sciences and Technology, Mahdia 5121, Tunisia; alibi_sana@hotmail.fr; 2BIOMEDAGE Group, Faculty of Medicine, Cantabria University, 39011 Santander, Spain; crespod@unican.es; 3Instituto de Investigación Valdecilla (IDIVAL), 39011 Santander, Spain

**Keywords:** small molecules, plant-derivates, antibacterial activity, multi-drug resistance

## Abstract

The vegetal world constitutes the main factory of chemical products, in particular secondary metabolites like phenols, phenolic acids, terpenoids, and alkaloids. Many of these compounds are small molecules with antibacterial activity, although very few are actually in the market as antibiotics for clinical practice or as food preservers. The path from the detection of antibacterial activity in a plant extract to the practical application of the active(s) compound(s) is long, and goes through their identification, purification, in vitro and in vivo analysis of their biological and pharmacological properties, and validation in clinical trials. This review presents an update of the main contributions published on the subject, focusing on the compounds that showed activity against multidrug-resistant relevant bacterial human pathogens, paying attention to their mechanisms of action and synergism with classical antibiotics.

## 1. Introduction

There is a wide range of plant species on Earth (400,000–500,000 species). Each plant is a small factory of chemical products, notably secondary metabolites, small molecules such as phenolics and polyphenols, terpenoids and essential oils, alkaloids, and others. Therefore, the vegetal world can be considered to be the largest manufacturer of compounds with biological activity, on earth. Many of these compounds show antimicrobial activity, although they do not receive the attention they deserve from doctors or veterinarians, because almost all antibiotics and antivirals on the market are produced by bacteria and fungi or are chemically synthesized. However, the use of plants as therapies against all kinds of ills is a social trend throughout the world, since the 90s of the XX century.

A considerable number of drugs in use today come from nature, either the microbial world, plants, or animals. The discovery and development of new compounds with pharmacological activity is largely based on the natural world. Approximately 75% of compounds with antibacterial activity introduced into clinical practice in the last 40 years derive from natural products [1]. There is a large body of data describing antimicrobial properties of plant extracts and constituents. The majority of these studies utilize unfractionated extracts that usually show weak in vitro antimicrobial activity. These studies were rarely confirmed by means of in vivo assays. Therefore, the precise action mechanism(s) of the large majority of such phytochemicals are unknown.

Since they are sessile organisms, plants developed the ability to synthesize compounds with a wide variety of colors, odors, and functions, which allow them to interact with the environment, as well as defend themselves from the attack of microorganisms and parasites [2]. The development as antimicrobials of most of the compounds from the plant world is complicated, due to their weak activity or their pharmacokinetic properties (difficulty of absorption in the gastro-intestinal tract or rapid metabolism). However, improvement of their biological and pharmacological properties can be achieved through structural modifications.

Numerous articles reporting on the antimicrobial activity of extracts of many plants are published, although fewer publications refer to the activity of specific molecules and their mechanism of action. In this review, we focus on the antibacterial activity of small molecules of vegetal origin against human pathogenic bacteria, in particular multi-drug resistant clinical strains, describing their mechanism of action (when known), and highlighting their anti-virulence properties. It mainly includes contributions published in the last 20 years.

## 2. Plant Phenols

Phenols are the largest group of secondary metabolites in plants, encompassing compounds such as simple phenols, phenolic acids, quinones, flavonoids, coumarins, lignans, and others. Their structure includes one or more hydroxyl (-OH) groups directly attached to an aromatic ring [3]. They are synthesized mainly from L-phenylalanine and L-tyrosine, through the shikimate pathway. Plant phenolics are weak acids due to the presence of aromatic rings and the hydroxyl phenolic group. They are acknowledged as strong natural antioxidants playing a key role in a wide range of biological and pharmacological properties like antimicrobial, anti-inflammatory, anticancer, antiallergic, and many more [4]. The antimicrobial activity of plant secondary metabolites is associated to the -OH group(s) attached to the phenol ring. The aromaticity can also be responsible for this effect [4]. Numerous phenolic compounds possess antibacterial activity.

### 2.1. Simple Phenols

Catechol and pyrogallol are hydroxylated phenols shown to be toxic to microorganisms. Catechol has two -OH groups, whereas pyrogallol has three (Figure 1). Increased hydroxylation results in increased antibacterial activity [5]. Several studies revealed the antiseptic and antimicrobial properties of pyrogallol. Catechol and pyrogallol showed activity against several Gram-positive and Gram-negative bacteria causing periodontitis [6], pyrogallol-based compounds being more active. Simple phenols interact with sulfhydryl groups of microbial enzymes, leading to their inhibition [7]. Eugenol is a hydroxyphenyl propene (Figure 1). It is a pale yellow, oily liquid, present in the essential oils of several plants, especially clove, nutmeg, and cinnamon. It is used as a flavoring agent in both food and cosmetics. Eugenol shows excellent antimicrobial activity against a wide range of Gram-negative and Gram-positive bacteria, including multi-drug resistant clinical isolates [8].

Resveratrol (3,5,4′-trihydroxystilbene) (Figure 1) is a natural polyphenol that is member of the stilbene family [9]. It is found in the grapes used for winemaking (*Vitis vinifera*) and in many other plants like peanuts (*Arachis hypogea*), blueberries, and cranberries (*Vaccinium* spp.) [10]. It is a phytoalexin synthesized by plants, in response to damage produced by fungal invasion [11]. Resveratrol exits as cis or trans isomer, the trans isomer being the most abundant and more stable of the two isomeric forms. From the end of XX century resveratrol became a popular molecule due to its potential health benefits [12]. In addition to extensive investigations on several disorders like cancer and vascular diseases, resveratrol is studied for its antimicrobial properties. The antibacterial properties of resveratrol were recently revised [13]. Resveratrol showed inhibitory activity in vitro against a wide range of pathogens, including Gram-positive and Gram-negative bacteria. However, for many species, the concentrations required to inhibit bacterial growth are higher than 100 mg/L [13]. The expression of virulence traits, such as biofilm formation and bacterial motility is modified by resveratrol [13]. The bacterial susceptibility to traditional antibiotics can be increased in the presence of resveratrol [13]. In humans, after oral administration, resveratrol is well absorbed and rapidly metabolized [14]. Therefore, it cannot reach the plasma concentrations required for antibacterial activity. This circumstance limits this route of administration in clinical practice, considering the inhibitory concentrations required [15]. A trial to improve bioavailability through intravenous administration revealed that resveratrol is also rapidly metabolized [16] However, the required bactericidal concentrations can be achieved via topical administration. Thus, a positive effect of resveratrol on lesion reduction in human acne vulgaris was observed [17].

### 2.2. Phenolic Acids

Phenolic acids or phenol-carboxylic acids are a family of organic compounds with a structure including a phenolic ring and a carboxylic acid group [18] (Figure 2).

The simplest of the members of this family is benzoic acid (BA), an aromatic carboxylic acid with a carboxyl group attached to a phenyl ring. Benzoic acid, benzoates, and benzoic acid esters are found in most fruits, particularly in berries (particularly in blueberries), mushrooms, cinnamon, and clove. Cinnamic acid (CinA) [(E)--3-phenylprop-2-enoic acid)] is an unsaturated monocarboxylic acid encompassing an acrylic acid carrying a phenyl substituent at the 3-position. It occurs as both cis and trans isomer in nature, although the trans isomer is more common. It is obtained from cinnamon oil and is also found in Shea butter. It is a precursor for the synthesis of more complex phenolic compounds: *p*-coumaric acid (*p*-CA), (4-hydroxycinnamic acid), is one of the three hydroxyl derivatives of CinA that differ by the position of the hydroxy substitution of the phenyl group. Caffeic acid (CFA), (E)-3-(3,4-dihidroxifenil)-prop-2-enoic acid, is also a hydroxyl derivative of CinA in which the phenyl ring is substituted by -OH groups at positions 3 and 4. It exists in cis and trans configurations, although the latter is more common. CFA is found in all plants because it is a key intermediate in biosynthesis of lignin. Furthermore, it acts as building block of a variety of plant metabolites. Rosmarinic acid (RA) is an ester of CFA and 3,4-dihydroxyphenyllactic acid that includes five hydroxyl groups in its chemical structure. Chicoric acid (ChA), (2*R*,3*R*)-2,3-bis{[(*E*)-3-(3,4-dihydroxyphenyl)-prop-2-enoyl]oxy}butanedioic acid, is a tartaric acid ester of two CFA molecules. It possesses six hydroxyl groups in its structure. Its most abundant form in nature is L-chicoric acid.

Phenolic acids are used as food preservatives, due to their recognized antibacterial and antifungal activities, for example BA (E210). The antimicrobial potential of each compound is related to the number of -OH and methoxy (-OCH_3_) groups present in the structure [19]. Thanks to their character of weak acids, they can diffuse through the bacterial membrane and acidify the cytoplasm, leading to cell death. [20]. Consequently, pK_a_ and lipophilicity are key parameters in the initial assessment of their bactericidal properties [9]. For example, CFA has a propene side chain, which makes it much more hydrophobic than, for example, protocatechuic acid. CFA lipophilicity facilitates its transport across bacterial cell membrane [21].

Some phenolic acids showed an antimicrobial activity similar to antibiotics. CFA showed comparable activity to ampicillin against *Staphylococcus aureus* and *Escherichia coli*. Additionally, gallic acid exhibited a good effect against a large group of bacteria including *Enterococcus faecalis, Pseudomonas aeruginosa*, *S. aureus*, *Moraxella catarrhalis*, *Streptococcus agalactiae*, *Campylobacter* sp., *Listeria monocytogenes*, and *Streptococcus pneumoniae* [22]. In another study, many phenolic acids including protocatechuic acid, 4-hydroxybenzoic acid, vanillic acid, syringic acid, CFA, p-coumaric, and ferulic acids contained in wild Polish mushrooms, showed intermediate antibacterial activities against a panel of Gram-positive and Gram-negative bacteria [23].

Propolis is a sticky, greenish-brown substance produced by bees and used as a coating to build their hives. Propolis contains more than 300 compounds, including phenolic acids and phenolic aldehydes, as well as flavonoids and quinones. Based on its composition, it is thought that propolis has antimicrobial properties. Ethanol extracts of propolis collected from apiaries located in different Polish regions showed moderate activity against *S. aureus* [24]. In addition, the extracts eradicated staphylococcal biofilm. When some of the extracts were combined with antibiotics inhibitors of protein synthesis (amikacin, kanamycin, gentamycin, tetracycline, and fusidic acid), a synergistic anti-staphylococcal effect was observed.

Hydrochloric extracts of *Pomegranate granatum,* which are rich in phenolics and anthocyanins, showed activity against Gram-positive and Gram-negative bacteria, including *P. aeruginosa* and *S. aureus*, with MIC values ranging between 30 and >90 mg/L [25].

Methyl gallate from *Galla rhois* originated from Asia exhibited strong activity against *Salmonella* sp. [26]. Synergistic interaction between methyl gallate and tylosine against *S*. *typhimurium* was reported [27]. This combination is a promising option to combat *S*. *typhimurium* in swine and, indirectly, safeguard the health of the public.

Hyperoside, catechin, CFA, ferulic acid, rutin, and ellagic acid found in extracts from 3 *Potentilla* species showed good antibacterial activity against Gram-positive and Gram-negative bacteria [28].

Pentagalloylglucopyranose contained in the extract of the seed kernels of *Mangifera indica* displayed a synergism with penicillin G against methicillin-resistant *S. aureus* (MRSA) [29].

The gallic acid potentially enhanced the activity of sulfamethoxazole, streptomycin, and tetracycline against several *P*. *aeruginosa* isolates, by altering the cell wall integrity [30].

### 2.3. Quinones

Quinones possess a fully conjugated cyclic dione structure [31] (Figure 3). In general, they resulted from the oxidation of hydroquinones [32]. Hydroxylated amino acids might be converted into quinones, in the presence of suitable enzymes.

These compounds are omnipresent in the environment—plants, fungi, lichens, bacteria, algae, viruses, insects, and higher organisms [31]. The carbonyl functions existing in quinones are related to the redox properties and are responsible for their activities [32]. Quinones are highly reactive electrophilic molecules that can covalently modify a variety of cellular nucleophiles. They react with functional groups of proteins or amino acids such as sulfhydryl, amine, amide, indole, and imidazole substituents. Primary products are Schiff bases, which are transformed to N-quinonyl derivatives or S-quinonyl derivatives, by 1,4-Michael addition via nitrogen or sulfur and to aldehydes in Strecker degradation [33]. The browning reaction of fruits and vegetables caused by cut or injury is the consequence of the formation of quinone-amino acids conjugates, which are colored. [34]. The same mechanism explains the staining propriety of henna [35]. Quinones also participate in the melanin synthesis process in human skin [34].

The majority of quinones found in plants are relatively simple benzoquinones, naphthoquinones or anthraquinones, although less common skeletal structures are also found to occur, such as terpenoid quinones and higher polycyclic quinones [36]. The benzoquinones are groups of compounds containing two carbonyl groups on a saturated hexacyclic aromatic ring system. The naphthoquinones and anthraquinones derived from fungi and higher plants. The naphthoquinones are characterized by their naphthalene nucleus with two carbonyl groups on one nucleus. While anthraquinones contain the anthracene nucleus with two carbonyl groups, polyquinones are dimers of the different types of quinones. Some of them are of mixed origin.

Quinones possess great antimicrobial activity. These compounds are able to form irreversible complexes with amino acids in proteins, which lead to their inactivation. Thus, they target cell wall constituents including surface-exposed adhesins, cell wall polypeptides, and membrane-bound enzymes [37]. They can also make substrates elusive to the microorganisms. In particular, 1,4-naphthoquinones show a great antibacterial potency and are promising agents for the treatment of bacterial infections.

Ravichandiran et al. [38] demonstrated that synthesized naphthoquinones have a good antibacterial effect against *S. aureus*, *L. monocytogenes*, *E. coli*, *P. aeruginosa*, and *Klebsiella pneumoniae*, with MICs values ranging between 15.6 and 500 mg/L. Some naphthoquinones extracted from *Newbouldia laevis* showed good activity against *Bacillus subtilis*, and *E. coli*. Unfortunately, naphthoquinones are toxic for living organisms, which limits their medical application [32].

Emodin, which is a natural anthraquinone derivative, occurs in many widely used Chinese medicinal herbs. It possesses a wide spectrum of pharmacological properties [39]. This molecule might be a potent antibacterial agent against Gram-positive bacteria, including MRSA and mycobacteria [39]. However, emodin could also lead to hepatotoxicity, kidney toxicity, and reproductive toxicity, particularly in high doses and with long-term use.

Thymoquinone is an active principle of *Nigella sativa* seed. It exhibited a significant bactericidal activity against Gram-positive cocci. This compound was able to inhibit biofilm formation and to prevent the adhesion of bacteria to glass slides surface [40].

Two studies [41,42] revealed the bacteriostatic effect of anthraquinones for a wide range of bacteria such as *Bacillus anthracis, Corynebacterium pseudodiphtheriticum,* and *P. aeruginosa,* and bactericidal activity against *Pseudomonas pseudomallei*.

Quinones present in *Lawsonia inermis,* originated from Oman henna, demonstrated antibacterial activity against *P. aeruginosa* [43]. Henna leaves extracts were active against *S. aureus* and *E. coli* [44].

Sudhir et al. [45] demonstrated that thymoquinone, thymohydroquinones, and dithymoquinone from *Nigella sativa* seeds have a good potency against *S. aureus, Streptococcus mutans*, and *Streptococcus mitis*.

The seeds of *Annona squamosa* are also rich in quinones, and exhibited a good antimicrobial effect against *S. aureus*, *E. faecalis*, *Staphylococcus epidermidis*, *E. coli*, and *P. aeruginosa* [46].

### 2.4. Flavonoids

Flavonoids are naturally occurring phenolic compounds named as 2-phenyl-1-benzopyran-4-one and classified into several subclasses, according to their basic chemical structures, such as isoflavone, flavones, flavanols, flavonols, flavanones, flavanonols, catechins, and anthocyanidins [47] (Figure 4). Almost 5000 known flavonoids are components of a wide variety of edible plants, fruit and vegetables, bearing phenyl chromanone backbone (C6–C3–C6) with -OH substitution. Therefore, flavonoids constitute a huge group of bioactive substances with a low systemic toxicity. Some flavonoids possess antimicrobial activity against a wide array of microorganisms.

#### 2.4.1. Flavones

Flavones are omnipresent in photosynthetic cells. They are present in fruit, vegetables, nuts, seeds, stems, and flowers. One of its main functions is protective action against microbial attacks. Due to this protective role, flavones are widely used in traditional medicine to combat infectious diseases [48]. Flavones are used as efflux pump inhibitors, mainly in Gram-positive bacteria [49]. Two plant-derived compounds, the flavonolignan 5′-methoxyhydnocarpin-D, and the porphyrin pheo-phorbide, are potent inhibitors of the NorA multi-drug resistant (MDR) efflux pump in *S. aureus* [50]. They did not exhibit any activity by themselves, but potentiated growth inhibitory activity of the plant antibacterial alkaloid berberine [51], the fluoroquinolone norfloxacin [52], and the antiseptic benzalkonium chloride [53] against multidrug-resistant *S. aureus.*

Apigenin and luteolin, two of the most ubiquitous plant flavones, showed moderate activity (MIC = 500 mg/L) against clinical strains of 4 bacterial species [54]. They were more active against Gram-negative (*E. coli* and *P. aeruginosa*) than Gram-positive (*E. faecalis* and *S. aureus*) bacteria. The presence of -OH groups in the phenyl rings A (C-5, C-7) and B (C-3′, C-4′) did not influence the activity level of flavones. Equally, the presence and position of the sugar group in the flavone glycosides generally had no effect on the MIC values.

Acacetin (Figure 3) is an O-methylated flavone present in many plants and dietary sources with various biological properties, including antimicrobial activity [55]. Acacetin is present in, among others, *Tunera diffusa, Propolis, Dracocephalum moldavica, Betula pendula, Robinia pseudoacacia, Flos Chrysanthemi Indici, Chrysanthemum, Safflower, Calamintha* and *Linaria* species. Acacetin is widely reported for a broad range of antimicrobial activity. Acacetin at 64 mg/L showed synergistic effects with norfloxacin, reducing its minimum inhibitory concentration (MIC) by two-fold [56]. It showed pronounced antibacterial activity against clinically isolated MRSA [57], by inhibiting the activity of sortase A protein [58], and restrained inhibition against *E. faecalis, E. coli,* and *P. aeruginosa* [59]. It also reduced the growth of *Actinomyces naeslundii, Actinomyces israelii, Actinobacillus* sp., *Prevotella intermedia,* and *Porphyromonas gingivalis*, but without effects against *Aggregatibacter actinomicecomitans* [60].

#### 2.4.2. Catechins

Tea is produced principally in Asian countries from ground dried leaves or buds of the *Camelia sinensis* plant. It is considered that tea with the most highly beneficial active ingredients for human health is green tea. Polyphenols are the biologically active components of green tea. Among them, the most medically relevant are the catechins and the theaflavins. The term catechin is commonly used to refer to the subgroup of flavan-3-ols (or simply flavanols) of flavonoids. The name of catechin comes from the family of plants called catechu (*Terra Japonica*) and specifically from the juice extracted from the Mimosa catechu (*Acacia catechu*). There are four main catechins in green tea: (–)-epicatechin (EC), (-)-epigallocatechin, (EGC), epicatechin-3-gallate (ECG), and (-)-epigallocatechin-3-gallate (EGCG). Four theaflavins [theaflavin (TF), theaflavin-3-gallate (TF3G), theaflavin-3′-gallate (TF3′G), and theaflavin-3,3′-digallate (TF33′G)] are formed by oxidative dimerization of catechins after harvest [61] (Figure 5).

Teas possess a broad-spectrum antimicrobial activity, but their precise limits are difficult to define, due to variation in the method of testing used. Antibacterial effects of tea were demonstrated against *S. aureus*, *E. coli*, *Vibrio cholerae*, *Shigella* spp., *Salmonella* spp., *Klebsiella* spp., *P. aeruginosa*, and *Bacillus* spp. [62]. However, they are only effective at very high MICs (0.5–1 g/L) that are unattainable in vivo, as a consequence of their weak antibacterial activity. Galloylated catechins, at moderate concentrations (3 mg/L) are able to phenotypically transform MRSA from full beta-lactam resistance (MIC = 256–512 mg/L) to complete susceptibility (MIC = 1 mg/L) [63]. Reversible conversion to susceptibility is mediated by a complex mechanism after intercalation of these compounds into the bacterial cytoplasm membrane, eliciting dispersal of the proteins associated with cell wall peptidoglycan synthesis, in the presence of beta-lactam antibiotics [62]. The use of these molecules in combination with suitable beta-lactam agents to treat MRSA infections seems feasible.

The relationships between structure and antibacterial activities of green tea catechins were recently reviewed [64]. There are 6 main antibacterial mechanisms—(i) inhibition of virulence factors (toxins and extracellular matrix); (ii) cell wall and cell membrane disruption; (iii) inhibition of intracellular enzymes; (iv) oxidative stress; (v) DNA damage; and (vi) iron chelation. The highest antibacterial activity was observed for galloylated compounds (EGCG, ECG, and theaflavin digallate). Catechins share a common binding mode, characterized by the formation of hydrogen bonds and hydrophobic interactions with their target.

EGCG, the major polyphenol of green tea, strongly inhibits the formation of certain kinds of biofilm, principally those that use amyloid fibers and pEtN-cellulose as principal extracellular matrix components [65].

### 2.5. Tannins

Tannins are complex polymers or oligomers found in almost all plant organs [66]. They regroup hydrolysable tannins (HTs), which are generally multiple esters of gallic acid with a carbohydrate (normally glucose), and condensed tannins or proanthocyanidins, which are derived from flavonoid monomers. Tannins might also be formed by the polymerization of quinone units [2].

HTs can be roughly grouped into two main categories—gallotannins and ellagitannins. They are formed by esterification of gallic acid or ellagic acids with a hydroxyl group of glucose. The antimicrobial activities of HTs were associated with the hydrolysis of the ester bond after ripening of many edible fruits [67]. HTs also show considerable synergy with antibiotics. However, they have low pharmacokinetic property [68].

Proanthocyanidins (PAs) are condensed tannins with several pharmacological properties, including antibacterial activity [69]. PAs are abundantly available in flowers, fruits, nuts, bark, and seeds [69]. They constitute a molecular defense of plants against biotic and abiotic stressors. They are oligomers and polymers made of catechin and epicatechin as building blocks. Berries are very rich in PAs, in particular lingonberry, cranberry, black currant, and blueberry [70]. The astringency of some unripe fruits, such as persimmon, banana, carob bean, and Chinese quince, is due to their high content in PAs. Medlar, plum, apricot, walnut, and pomegranate are other fruits with high PA content [71].

PA dimers are found to be potent antimicrobial agents [72]. Different PAs isolated from the stems of *Ephedra sinica* showed antibacterial activity against either *P. aeruginosa* (MIC = 0.0835 mM) or MRSA (MIC = 0.0817 mM) [73]. The viable *L. monocytogenes* cell count were reduced by means of a grape seed extract rich in PAs [74]. PAs exhibited anti-adherence property towards MDR strains of uropathogenic P-fimbriae-expressing *E. coli* [75]. Cranberry extract rich in PAs reduced virulence and inhibited quorum sensing of *P. aeruginosa* in the model host *Drosophila melanogaster* [76]. PA trimer present in the peanut skin disrupts cell wall integrity of *Bacillus cereus* [77]. Ellura, a proanthocyanidin-rich commercial product from cranberry, is commonly used for prevention of urinary tract infections (UTIs) [78].

## 3. Terpenoids

Most plants synthesize and release terpenes, volatile compounds that perform various ecological functions and have significant environmental impact [79]. They constitute a family of low molecular weight compounds with an important influence on human life, due to their applications in medicine, food, and cosmetics. Essential oils are terpene-related secondary metabolites responsible of the fragrance of plants [80]. Terpenes are polymers of isoprene (C_5_H_8_) and can be classified in 8 classes—hemiterpenes, monoterpenes, sesquiterpenes, diterpenes, sesterterpenes, triterpenes, tetraterpenes, and polyterpenes, which differ in the number of isoprene units. Terpenoids are synthesized from acetate units and contain additional elements (Figure 6). Several studies reported the antibacterial effects of terpenoids, including mono-, di-, and triterpenes [81,82].

The antibacterial mode of action of most terpenoids is related with alterations of the photorespiratory pathway and the photosynthetic machinery caused by the inhibition of the glutamate and aspartate metabolism [83]. Their antibacterial activity was related to— (i) the presence of hydroxyl groups; (ii) lipophilicity/hydrophobicity; and (iii) carbonylation of terpenoids [84].

Twenty-five constituents of essential oils were evaluated against *Mycobacterium tuberculosis* and *Mycobacterium bovis* [85] by the Alamar Blue technique. Carvacrol and thymol were the most active terpenes, with MIC values of 2.02 and 0.78 mg/L, respectively.

Carvacrol or 2-methyl-5-(1-methylethyl)-phenol (Figure 6) is a monoterpenic phenol, biosynthesized from γ-terpinene through p-cymene. Carvacrol occurs in aromatic plants and in many essential oils of the *Labiatae* family, including oregano (*Origanum vulgare*), thyme (*Thymus vulgaris*), pepperwort (*Lepidium flavum*), wild bergamot (*Citrus aurantium bergamia*), and others. It is approved as a safe food additive in the USA and Europe and is used as a flavoring agent in sweets, beverages, and chewing gum. Medicinal plants containing carvacrol were used in folk medicine for a long time, even before studies on their therapeutic effectiveness became known. Carvacrol is reported to have a wide variety of biological properties including antimicrobial activity. In this context, two reviews on the most relevant aspect of these activities were previously reported [86,87]. Compared to other volatile compounds present in essential oils, carvacrol shows higher antimicrobial power, due to the phenol ring, which confers hydrophobicity, and the presence of the free hydroxyl group. Carvacrol is active against a huge variety of Gram-positive and Gram-negative human pathogenic bacteria, encompassing both planktonic and sessile human pathogens. In particular, it is very effective to control food-borne pathogens, such as *E. coli*, *Salmonella*, and *Bacillus cereus*. Moreover, carvacrol lends itself to being combined with nanomaterials, thus providing an opportunity for the development of new anti-infective substances to prevent biofilm-associated infections [88].

Thymol (2-isopropyl-5-methylphenol) is a carvacrol isomer also known as “hydroxy cymene”. As for carvacrol, natural sources of thymol are the essential oils extracted from plants of the *Labiatae* family, such as thyme, oregano, etc. The antibacterial activities of thymol were described in detail in a recent review [89]. Thymol possesses antibacterial activity against a wide range of species, including biofilm-embedded microorganisms. However, the practical application of thymol as antibiotic is limited by the high dosage required for bactericide action as well as its toxicity for animal cells. Therefore, specific studies are required to verify if optimal antimicrobial activity can be achieved using non-toxic concentrations.

## 4. Alkaloids

Alkaloids are a large and structurally diverse group of secondary metabolites generally synthesized from amino acids, which have in common their basic nature, which allows them to be found either as salts of organic acid or as free bases. Alkaloids are water-soluble at acid pH and are soluble in organic solvents at alkaline pH. The natural sources of alkaloids are plants (they can be found in 300 plant families) as well as bacteria, fungi, and animals [90]. Some plants use them as natural insecticides/pesticides to defend the plant from the damaging action of some insect species. Vegetal alkaloids are mainly synthesized in vascular and herbaceous plants. Alkaloids are unevenly distributed in the plant tissues, some are highly concentrated in the seeds, while other are found in leaves, barks, fruits, and even roots. Different parts of the same plant might contain different types of alkaloids. More than 18,000 alkaloids are discovered to date [91]. Alkaloids are a structurally diverse group of molecules with a unique unifying feature, which is the presence of a basic nitrogen atom that is part of an amine group [a primary amine (RNH_2_), a secondary amine (R_2_NH) or a tertiary amine (R_3_N)] [92]. They can appear as monomeric forms or they can form oligomers. Alkaloids can generally be categorized on the basis of their chemical structure, metabolic pathway, or natural origin, although there is no standard taxonomic classification [93].

Different biosynthesis pathways drive to three major categories of alkaloids: true-, proto-, and pseudo-alkaloids. Amino acids are the precursors of true- and proto-alkaloids, but they differ in terms of the presence of the N-atom in the heterocycle, respectively. Pseudoalkaloids possess a basic carbon skeleton not derived from an amino acid [94]. Due to their structural complexity and according to their backbone, heterocyclic alkaloids can be divided into 14 subgroups, including indoles, isoquinolines, pyrrolidines, pyrrolizidines, quinolizidines, tropanes, purines, piperidines, and imidazoles [94]. Alkaloids have a wide variety of pharmacological actions, including antibacterial activity. Alkaloids have a proton accepting nitrogen and, on some occasions, several proton-donating amine hydrogens. Therefore, their biological activity is suggested to be related to the property of forming hydrogen bonds with several biomolecules, such as receptors, enzymes, and proteins [95]. As an example, *Datura stramonium* is a well-known plant that contains several alkaloids showing antimicrobial activity against several microorganisms, among them *E. coli* and *P. aeruginosa* [96].

The antibacterial properties of plant alkaloids were extensively reread in two recent reviews [97,98]. Here, we focus on alkaloids with antibacterial activity specific for multidrug-resistant pathogens.

Caffeine (Figure 7) is a xanthine alkaloid found in the seeds, nuts, or leaves of a number of plants native to Africa, East Asia, and South America, and helps to protect them against herbivory and from competition, by preventing the germination of nearby seeds [99]. The most well-known source of caffeine is the coffee bean, the seed of the *Coffea* plant. The antibacterial activity of coffee extracts against enterobacteria was previously reported [100]. *P. aeruginosa* is a potent biofilm forming organism causing several diseases. Caffeine showed substantial activity against *P. aeruginosa* (MIC = 200 mg/L), and significantly inhibited its biofilm development [100]. Caffeine interferes with the quorum sensing of *P. aeruginosa* by targeting the swarming motility. Molecular docking analysis indicated that caffeine might interact with the quorum sensing proteins, namely LasR and LasI. Thus, caffeine could inhibit the formation of biofilm by interfering with the quorum sensing of the organism. Caffeine is also found to reduce the secretion of virulence factors from *P. aeruginosa* [100].

Berberine is a quaternary ammonium salt from the protoberberine group of benzylisoquinoline alkaloids (Figure 7) found in the roots of plants like *Berberis vulgaris*, *Argemone mexicana*, *Coptis chinensis*, *Mahonia aquifolium*, and others [101]. Berberine is widely used in the clinical treatment of bacterial diarrhea [102]. The mechanism of action of berberine on diarrhea caused by *V. cholerae* and *E. coli* was extensively studied. As early as 1982, berberine was shown to directly inhibit *V. cholerae* and *E. coli* enterotoxins effects in vitro [103]. In *Streptococcus pyogenes*, berberine sulfate interferes with the process of fimbrial formation, inhibiting bacterial adherence to the mucosal or epithelial surfaces [104]. It is difficult to explain the effects of berberine, as although it is a cationic alkaloid, it is poorly soluble in water, which implies a reduced intestinal absorption (in animal models it was observed that the bioavailability of berberine was much lesser than 1%, since that part of the berberine absorbed by the intestine is excreted back into the intestinal lumen through the action of the P-glycoprotein). Berberine also shows antibacterial effect in vitro against *S. aureus* [105]. A combination of berberine and vancomycin was more effective than vancomycin alone for treating *Clostridium difficile* infection in a mice model [106]. While many of the effects of berberine are explained through enzyme / receptor and cell signaling mechanisms, there is growing evidence that these effects are due to an interrelationship between berberine and the gut microbiota, similar to those of probiotics and dietary fiber [107].

Berberine and CinA can self-assemble, forming nanoparticles (NPs) that display bacteriostatic activity against MRSA and show ability for biofilm elimination [108]. CinA is one of the main components of the traditional spice *Cinnamomi cortex*, widely used in daily life [109]. Berberine-CinA NPs can first spontaneously adhere to the bacterial surface, penetrate into the cell, and then lead to converging attack against MRSA. Bactericidal mechanisms of Berberine-CinA NPs on MRSA were revealed by both transcriptomic and quantitative Polymerase Chain Reaction analysis. Combining the results of these analysis with spectral data and single crystal X-ray diffraction, allowed elucidation of NPs’ self-assembly mechanism, which was driven by hydrogen bonds and π−π stacking interactions. Biocompatibility tests showed that the NPs are nonhemolytic, with little toxicity in vitro and in vivo. This directed self-assembly mode offers a new perspective toward the development of biocompatible plant-based antimicrobial nanomedicines, with potential clinical application.

Capsaicin (CAP) (8-methyl-N-vanillyl-6-nonenamide) is an alkaloid found in the berries of almost all peppers (*Capsicum* sp.) [110]. CAP is one of the most used spices in cuisines all over the world [111]. In addition to its culinary utilization, CAP is used for pain relief in several serious chronic conditions [112,113].

Among its multiple benefits for human health, CAP possess antibacterial properties. At concentrations of 12.5–50 mg/L, CAP inhibits the *S. aureus* NorA efflux pump, reducing its susceptibility to ciprofloxacin (the MIC was reduced by 2- to 4-fold in presence of capsaicin) and the invasiveness of *S. aureus* in the J774 macrophage cell line [114]. Using a mice virulence model, Qiu et al. demonstrated that capsaicin (dosage 100 mg/Kg) protects mice from pneumonia caused by MRSA [115]. CAP showed moderate bactericide activity (MIC = 64–128 mg/L) against a collection of 32 *Streptococcus pyogenes* Italian clinical isolates from children with pharyngitis [116]. At sublethal concentrations (8–32 mg/L), capsaicin prevented cell invasion and reduced hemolytic activity, two important virulence features of this pathogen.

## 5. Conclusions

The emergence of antibiotic-resistant microorganisms is a global health threat that limits successful therapeutic management of bacterial infections. It is now essential to develop new strategies to fight antibiotic resistance. The usage of natural compounds derived from plants attracted the attention of multi-disciplinary researchers. These compounds exert their antibacterial activities by means of a variety of mechanisms (Figure 8). Some phytochemicals reported in this review are already in use as food preservatives. Nevertheless, most plant-derived molecules are not suitable for development as anti-infective agents for clinical use, because they are effective against important bacterial pathogens, only at concentrations that are unobtainable in vivo. However, some of them show synergistic actions with classical antibiotics. These actions turn them into molecules with a promising future, to be applied in the clinic. The capacity showed by tea galloylated catechins to fully reverse β-lactam resistance in MRSA at low concentrations is remarkable and pave the way for a new generation of compounds that can resolve infections, not due to their antibacterial properties reflected in the MIC but due to their ability to counteract drug resistance machineries. This is an example of how plant-derived compounds can preserve the efficacy of old antibiotics, and consequently restore their clinical application.

Some compounds showing promising antibacterial properties, such as the mentioned catechins and resveratrol, are rapidly metabolized once they are supplied by the common routes. Their structures can be chemically modified to obtain more acceptable compounds from a pharmaceutical perspective.

Phytochemicals that are characterized by diverse chemical structures and mechanisms of action are attractive therapeutic tools for discovering new active products. A large amount of information on the antibacterial properties of plant extracts and active molecules has been published so far in this century. Many studies are still needed to shed light on the antibacterial mechanisms and pharmacokinetic properties of plant-derived molecules. It would also be desirable to unify the methods and units used to assess the antibacterial activity of these compounds. For the most promising compounds, the translation of in vitro studies to in vivo experiments and finally to human clinical trials need to be conducted.

## Figures and Tables

**Figure 1 antibiotics-10-00231-f001:**
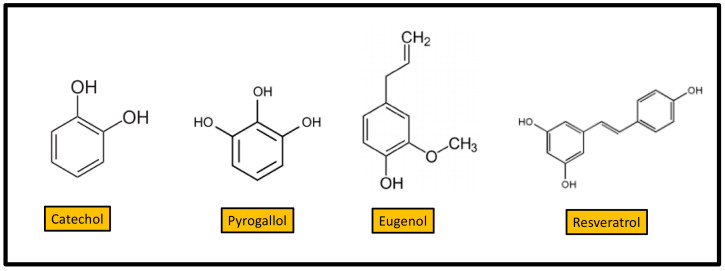
Structure of four simple phenols with antibacterial activity.

**Figure 2 antibiotics-10-00231-f002:**
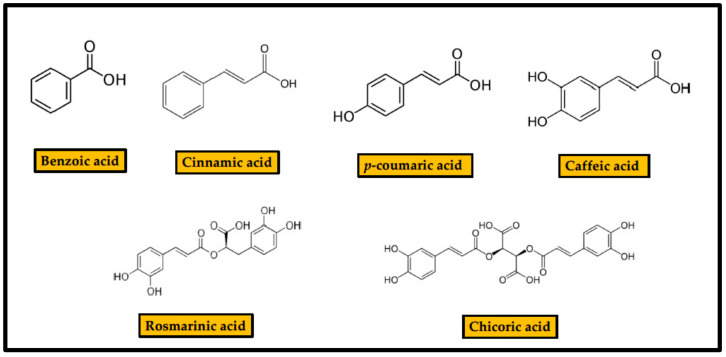
Structure of phenolic acids.

**Figure 3 antibiotics-10-00231-f003:**
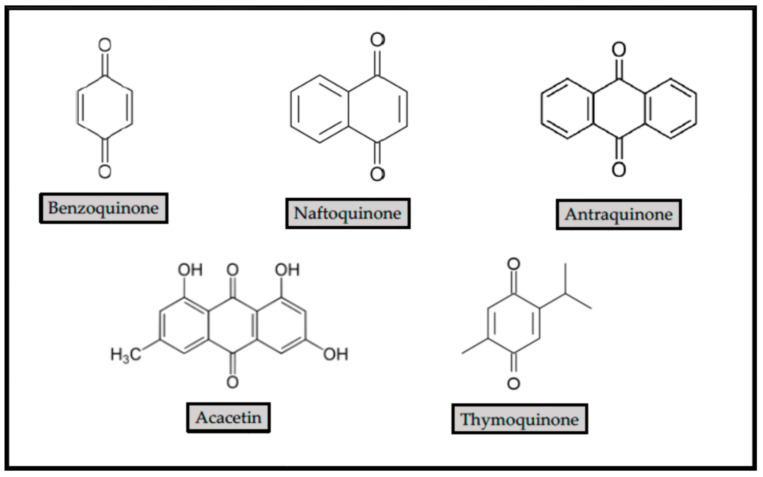
Structure of main quinone groups, and two quinones with antibacterial properties, acacetin and thymoquinone.

**Figure 4 antibiotics-10-00231-f004:**
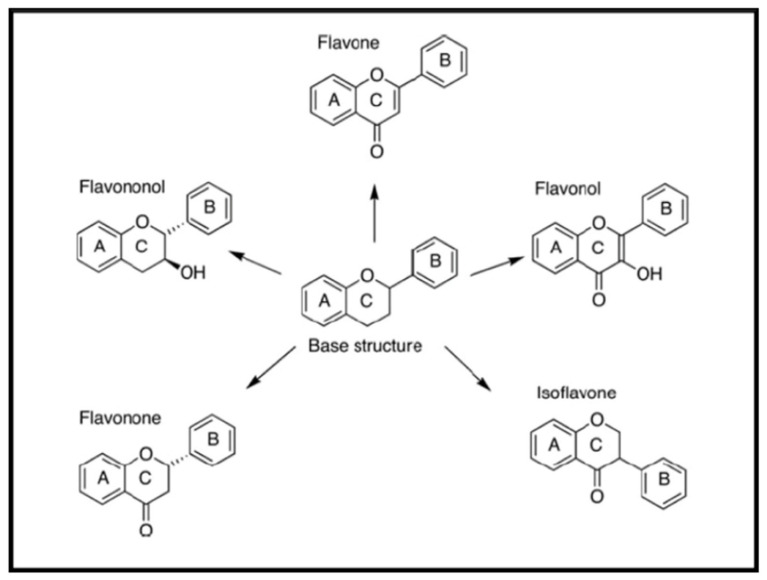
Structure of flavonoids.

**Figure 5 antibiotics-10-00231-f005:**
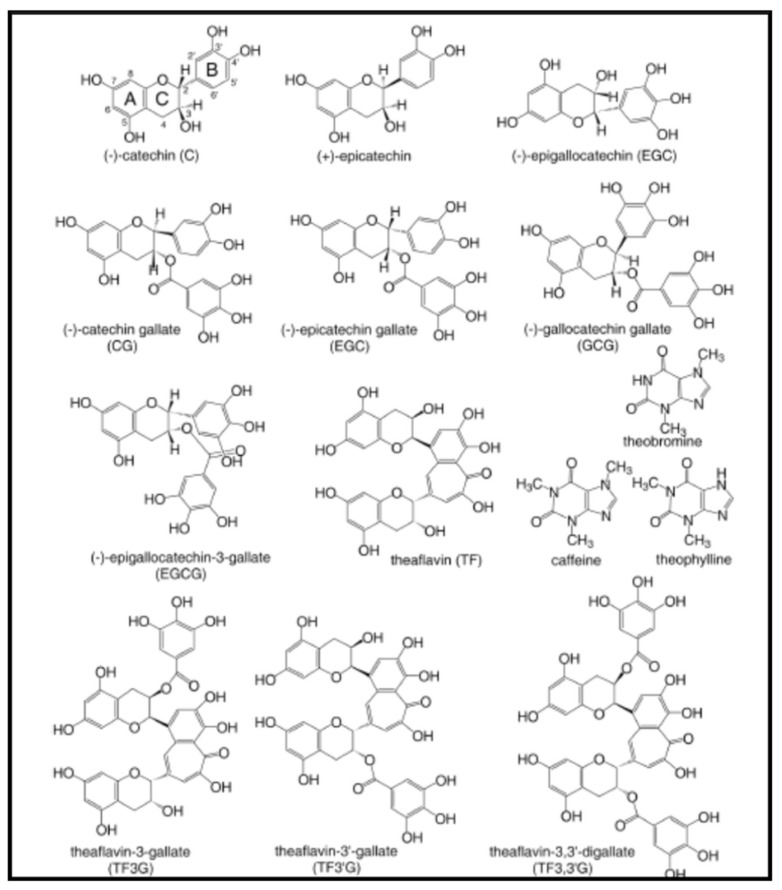
Structure of tea catechins.

**Figure 6 antibiotics-10-00231-f006:**
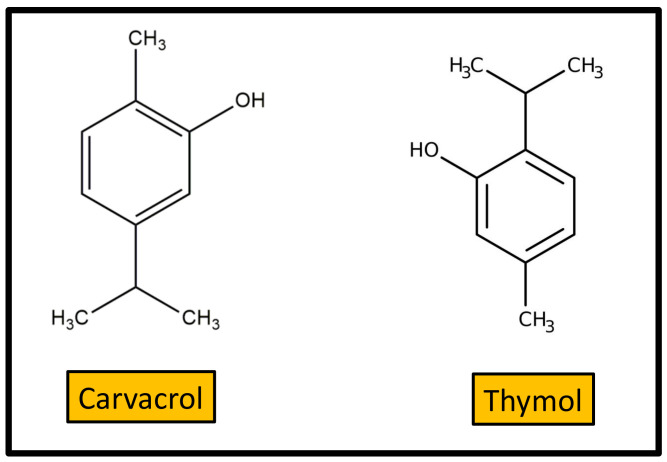
Structure of carvacrol and its isomer thymol.

**Figure 7 antibiotics-10-00231-f007:**
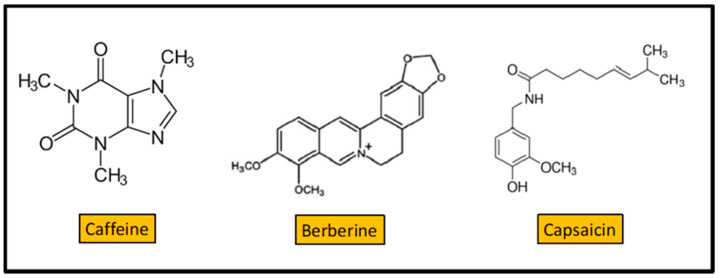
Structure of 3 alkaloids: caffeine, berberine, and capsaicin.

**Figure 8 antibiotics-10-00231-f008:**
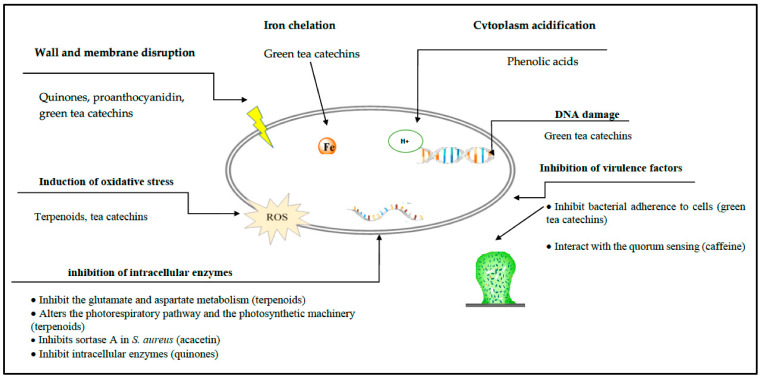
Antibacterial action mechanisms of plant-derived compounds.

## Data Availability

No new data were created or analyzed in this study. Data sharing is not applicable to this article.

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
