# Peer review of "Plant-Derivatives Small Molecules with Antibacterial Activity"

_antibiotics, 2021, doi:10.3390/antibiotics10030231_

Round 1

Reviewer 1 Report

The present review reports an update of the main contributions published in the field of natural product chemistry on compounds endowed with activity against multidrug-resistant relevant bacterial human pathogens. A special attention is paid on their mechanisms of action and synergism with classical antibiotics.

The review is well-written and logically-structured.

Although many papers already reports the antimicrobial activity of “in toto” extracts, the merit of this review is to focus on the activity of specific molecules and their mechanism of action.

- Seven figures have been provided on the chemistry of selected small molecules object of the discussion. However, none of them reports on mechanisms of action against pathogens. 

- No time frame concerning the literature search is reported. 

Author Response

The present review reports an update of the main contributions published in the field of natural product chemistry on compounds endowed with activity against multidrug-resistant relevant bacterial human pathogens. A special attention is paid on their mechanisms of action and synergism with classical antibiotics.

The review is well-written and logically-structured.

Although many papers already reports the antimicrobial activity of “in toto” extracts, the merit of this review is to focus on the activity of specific molecules and their mechanism of action.

- Seven figures have been provided on the chemistry of selected small molecules object of the discussion. However, none of them reports on mechanisms of action against pathogens. 

A new figure has been introduced (Figure 8) showing the main antibacterial mechanisms of plan-derived compounds

- No time frame concerning the literature search is reported. 

This sentence has been included in the INTRODUCTION section: “It mainly includes contributions published in the last 20 years”. In the CONCLUSIONS section this sentence has been added: “A huge amount of information on the antibacterial properties of plant extracts and active molecules has been published so far this century”.  

Reviewer 2 Report

  1. Moderate corrections of English language are necessary through the whole manuscript.
  2. Some sentences need to be rewritten. For example:
    1. This circumstance limits this via of administration for systemic use of resveratrol to treat bacterial infections
    2. Among its many functions include its protective action against microbial
    3. Main focus should be giving also to the mechanism of action of plant derivates, since this knowledge facilitates the interpretation of results then their application for chemotherapeutic treatment of infections.
  3. „Most of the drugs in use today come from nature either of the microbial world, plants or animals.“ This statement is not correct.
  4. Structures of the compounds in Figures are not clear enough, and often blurry.
  5. What is the mechanism of formation of irreversible complexes of quinones and amino acids?
  6. Cinnamic acid is mentioned 2 times in the manuscript: page 3, abbreviation CinA, and on Page 12, abbreviation CA. It seems that 2 authors wrote that paragraphs.
  7. Conclusions are not strong enough. Please rewrite.

In general, it is a nice manuscript and I recommend acceptance after minor revision.

Author Response

  1. Moderate corrections of English language are necessary through the whole manuscript.
  2. Some sentences need to be rewritten. For example:

    1. This circumstance limits this via of administration for systemic use of resveratrol to treat bacterial infections

This sentence has been rewritten: “This circumstance limits this route of administration in clinical practice, considering the inhibitory concentrations required”.

    1. Among its many functions include its protective action against microbial

“One of its main functions is protective action against microbial attacks”

    1. Main focus should be giving also to the mechanism of action of plant derivates, since this knowledge facilitates the interpretation of results then their application for chemotherapeutic treatment of infections.

This sentence has been eliminated, since CONCLUSION section has been fully rewritten, as suggested.

  1. „Most of the drugs in use today come from nature either of the microbial world, plants or animals.“ This statement is not correct.

This sentence has been modify: “A considerable number of drugs in use today come from nature either of the microbial world, plants or animals”.

  1. Structures of the compounds in Figures are not clear enough, and often blurry.

Now the figures in pdf manuscript are sharper, at least in my computer I can see them sharp. We can try to improve the quality of figures following the instructions of the editing services of the journal, if the manuscript is accepted.

  1. What is the mechanism of formation of irreversible complexes of quinones and amino acids?

The mechanism is described in detail (L171-184). A new reference has been introduced.

  1. Cinnamic acid is mentioned 2 times in the manuscript: page 3, abbreviation CinA, and on Page 12, abbreviation CA. It seems that 2 authors wrote that paragraphs.

CinA is the abbreviation used in all manuscript. The required modifications have been introduced (L427-432).

  1. Conclusions are not strong enough. Please rewrite.

Conclusions have been fully rewritten.

In general, it is a nice manuscript and I recommend acceptance after minor revision.

Reviewer 3 Report

The authors have reviewed antibacterial activity of plant ingredients.

In the discussion, it is desirable to add whether there is a promising antibacterial agent developed with the plant-derivative as a lead compound.

In conclusion, it is desirable to add about whether the ingredients have potential as antibacterial agents.

Author Response

The authors have reviewed antibacterial activity of plant ingredients.

In the discussion, it is desirable to add whether there is a promising antibacterial agent developed with the plant-derivative as a lead compound.

In conclusion, it is desirable to add about whether the ingredients have potential as antibacterial agents.

The majority of plant-derived molecules are not suitable for development as anti-infective agents for clinical use, because they are effective against important bacterial pathogens only at concentrations that are unobtainable in vivo. However, some of them have shown some kind of synergistic actions with classical antibiotics. For instance, the capacity showed by tea galloylated catechins at low concentrations to fully reverse beta-lactam resistance in methicillin-resistant Staphylococcus aureus is remarkable and pave the way to a new generation of compounds that can resolve infections due not to their antibacterial properties reflected in the MIC but to their ability to counteract drug resistance machineries.

This has been included in CONCLUSIONS, as well as a commentary on the possibility of improving the pharmacological properties of these compounds by modifying their chemical structure.

Round 2

Reviewer 3 Report

This review has been greatly improved.

Especially in the conclusion section, it's much better.

This manuscript is a resubmission of an earlier submission. The following is a list of the peer review reports and author responses from that submission.